# Relation between Vitamin D and COVID-19 in Aged People: A Systematic Review

**DOI:** 10.3390/nu13041339

**Published:** 2021-04-17

**Authors:** Moustapha Dramé, Cécilia Cofais, Maxime Hentzien, Emeline Proye, Pécory Souleymane Coulibaly, David Demoustier-Tampère, Marc-Henri Destailleur, Maxime Lotin, Eléonore Cantegrit, Agnès Cebille, Anne Desprez, Fanny Blondiau, Lukshe Kanagaratnam, Lidvine Godaert

**Affiliations:** 1Faculty of Medicine, University of the French West Indies, 97200 Fort-de-France, France; 2Department of Clinical Research and Innovation, University Hospitals of Martinique, Pierre Zobda-Quitman Hospital, 97261 Fort-de-France, France; 3Department of Geriatrics, University Hospitals of Rennes, 35000 Rennes, France; cecilia.cofais@chu-rennes.fr; 4Department of Infectious Diseases, University Hospitals of Reims, 51100 Reims, France; mhentzien@chu-reims.fr; 5Department of Geriatrics, General Hospital of Valenciennes, 59300 Valenciennes, France; proye-e@ch-valenciennes.fr (E.P.); coulibaly-ps@ch-valenciennes.fr (P.S.C.); demoustiertampere-d@ch-valenciennes.fr (D.D.-T.); destailleur-mh@ch-valenciennes.fr (M.-H.D.); lotin-m@ch-valenciennes.fr (M.L.); cantegrit-e@ch-valenciennes.fr (E.C.); cebille-a@ch-valenciennes.fr (A.C.); desprez-a@ch-valenciennes.fr (A.D.); blondiau-f@ch-valenciennes.fr (F.B.); 6Department of Clinical Research and Innovation, University Hospitals of Reims, 51100 Reims, France; lkanagaratnam@chu-reims.fr

**Keywords:** vitamin D, COVID-19, aged people

## Abstract

Background: Vitamin D has diverse and extensive effects on the immune system, including activating innate immunity and reducing the overactive adaptive immune response. A systematic review was performed to identify and synthesize the best available evidence on the association between vitamin D level and risk of COVID-19, adverse outcomes and possible benefits of supplementation in aged 60 years or over. Methods: A literature search was performed in PubMed© and Scopus© for all publications from inception published before 15 March 2021. Studies reporting data from aged patients on vitamin D use and COVID-19 were included. Basic science articles, editorials and correspondence were excluded. Publication year, study design and setting, characteristics of the study population were extracted. This study is registered with PROSPERO, under the number CRD42020223993. Results: In total, 707 studies were identified, of which 11 observational studies were included in the final review. Four studies compared vitamin D-supplemented COVID-19 patients to non-supplemented patients, and seven compared patients with vitamin D deficiency to patients without deficiency. In all four studies, patients with vitamin D supplementation had better rates of primary clinical outcomes (death, the severity of the disease, oxygen therapy requirement…). In studies comparing patients with vitamin D deficiency and patients without vitamin D deficiency, those without vitamin D deficiency had better primary clinical outcomes (death rate, the severity of the disease, oxygen therapy requirement, invasive mechanical ventilation need…). Conclusion: This systematic review seems to support an association between vitamin D deficiency and the risk of COVID-19 in aged people. In addition, vitamin D deficiency appears to expose these subjects to a greater risk of adverse outcomes. Because of its simplicity of administration, and the rarity of side effects, including vitamin D in preventive strategies for certain viral diseases, it appears to be an attractive option.

## 1. Introduction

Infection with severe acute respiratory syndrome coronavirus 2 (SARS-CoV-2), particularly COVID-19, has been at the origin of a worldwide pandemic since December 2019. While most infected individuals have mild to moderate signs and a spontaneously favorable course, some infected persons present severe forms of infection that can lead to death [1,2]. The severe form of COVID-19 infection is characterized by acute respiratory distress secondary to severe lung damage [3,4]. Respiratory distress most often occurs more than seven days after the onset of symptoms [5] and is frequently observed in aged people [6,7,8]. Several authors have demonstrated abnormally high levels of proinflammatory cytokines in patients developing severe forms of COVID-19 infection [9,10]. In light of this, it has been suggested that the respiratory complications of COVID-19 infection occurring beyond day seven were due to a dysregulation of the immune system, termed the “cytokine storm” [11]. One strategy to limit severe forms of COVID-19 would be to limit or prevent this cytokine storm. Activated vitamin D is involved at different immune response levels [12], notably by activating innate immunity and reducing overactivation of the adaptive immune system [13]. Vitamin D administration may lower interleukin-6 levels [10]. In 2009, Grant et al. [14] suggested a possible vitamin D role in reducing case-fatality rates from the 1918–1919 influenza pandemic. Conversely, vitamin D deficiency may contribute to deregulation of the immune system [13] and has been reported to be associated with a higher risk of intensive care admission and mortality in people with severe forms of pneumonia [12,15].

The objective of this work was to perform a systematic review to identify and synthesize the best available evidence on the association between vitamin D level and risk of COVID-19, adverse outcomes and possible benefits of supplementation aged 60 years or over.

## 2. Methods

The question to be answered by this systematic review was to determine whether there is any available evidence on the association between vitamin D deficiency (compared to non-vitamin D deficiency) or vitamin D supplementation (compared to non-vitamin D supplementation), and risk of COVID-19 or adverse outcome, in people aged 60 years or over.

### 2.1. Search Strategy

This was a systematic review only. A comprehensive literature search was performed using PubMed and Scopus. The search covered all publications up to and, including 5 November, 2020, with no specific start date specified. Search terms were defined by two senior researchers (LG, MD) and included the following keywords in the title and/or the abstract: (“vitamin d” OR calciferol OR calcitriol) AND (covid OR coronavirus OR SARS OR “cytokine storm” OR “respiratory infection” OR “acute respiratory distress syndrome”). Filters were applied to select studies in English language studies, including human beings only, and exclude the following publication types: reviews, case reports and case series, editorials, and correspondence. Additional studies were identified from reviewing the reference lists of retrieved studies. The authors of the studies were contacted to recover unpublished data when available. Study selection was performed following the PRISMA (Preferred Reporting Items for Systematic Reviews and Meta-Analyses) guidelines. This study is registered with PROSPERO, under the number CRD42020223993.

### 2.2. Study Selection Criteria

Study eligibility criteria were defined before performing the literature search by two senior researchers (LG, MD). Studies were eligible for inclusion if they reported data on vitamin D and infection and if the study population’s mean age was 60 years or over. Basic science articles, reviews, case reports and case series, editorials, and correspondence were excluded.

### 2.3. Data Extraction

Data analysis was performed using Covidence systematic review software© (Veritas Health Innovation, Melbourne, Australia) available at www.covidence.org (accessed on 8 April 2021). After eliminating duplicates, two senior researchers (LG, MD) independently reviewed the titles and abstracts of all articles (after they had been rendered anonymous). In case of disagreement about whether or not to include an article, the case was discussed until consensus was reached. Overlap between studies in the results reported was checked. They independently extracted the data using the same data extraction form. The following data were extracted: publication year, study design, study setting, characteristics of the study population, i.e., number of subjects included, the proportion of females, mean and/or median age, the proportion of COVID-19-positive patients in the study population, and type of comparison (comparison between patients supplemented in vitamin D and non-supplemented patients, or comparison between patients with vs. without vitamin D deficiency). In the supplementation studies, the type of vitamin D used, the supplementation regimen, and outcomes (death, invasive mechanical ventilation need, the severity of COVID-19…) were collected. In studies comparing patients with vs. without vitamin D deficiency, the following data were collected: serum vitamin D levels, percentage of patients with vitamin D deficiency, the definition used for vitamin D deficiency or insufficiency, and outcomes. When appropriate, some authors were contacted for data specific to the subpopulation of persons aged 60 years or older.

### 2.4. Quality Assessment

The quality of included studies was assessed independently by two researchers (LG MD) using the Newcastle–Ottawa scale (NOS) [16] for cohort studies and a modified version of the Newcastle–Ottawa scale for cross-sectional studies. The NOS consists of three quality parameters: selection, comparability, and outcome assessment assigning a maximum of four points (five points for cross-sectional studies) for selection, two points for comparability, and three points for the outcome. NOS scores of 7 or over were considered as high-quality, and of 5–6 as moderate quality. Disagreement was resolved by joint review of the manuscript to reach consensus, and the opinion of a third researcher was requested when necessary.

## 3. Results

In total, 707 studies were identified by the literature search (Figure 1). Among these, 276 duplicates were excluded. After examining the titles and abstracts of the remaining 432 studies, 112 articles were retained for full-text assessment. After reading the full text of these 112 studies, 101 were excluded for one or more of the following reasons: wrong study design, wrong study population, wrong outcome criterion, or overlapping data. Thus, 11 studies were included in the final review [17,18,19,20,21,22,23,24,25,26,27]. Because of the heterogeneity of outcomes between studies, a meta-analysis was not performed. Some authors agreed to provide data specific to the subpopulation of persons aged 60 years or older [19,20,21,22,23,24,25,26]. Concerning the study by Tan et al. [27], the systematic review data were restricted to the subgroup of 60 years or older.

Table 1 summarizes the characteristics of the studies included in the review. All studies were observational, seven were retrospective [17,18,20,22,23,24,26], and four were prospective [19,21,25,27]. Four studies compared vitamin D-supplemented patients to non-supplemented patients [17,18,22,27], while seven compared patients with vs. without vitamin D deficiency [19,20,21,23,24,25,26]. Different adverse disease courses were used as single or composite clinical outcome measures: death, ICU support, oxygen therapy requirement, or invasive mechanical ventilation need. Serum vitamin D level was significantly higher in COVID-19-negative patients compared to COVID-19-positive patients in the study by Baktash et al. [19] (20.8 ng/mL vs.10.8 ng/mL; *p* = 0.008), as well as in the study by Sulli et al. [26] (16.3 ng/mL vs. 7.9 ng/mL; *p* = 0.001).

The different cutoffs used in the studies to define vitamin D groups (insufficiency, deficiency, or severe deficiency) are described in Appendix A.

Table 2 presents the vitamin D supplementation regimen, outcomes criteria, and results. In all four studies, patients with vitamin D supplementation had better clinical outcomes.

For studies on vitamin D deficiency, Table 3 presents the description of vitamin D status (serum vitamin D levels, vitamin D deficiency rates) of elderly patients with and/or without COVID-19. In these studies, patients without vitamin D deficiency had better clinical outcomes than patients with vitamin D deficiency (Table 4).

Study quality as assessed using the NOS is summarized in Table 5. The quality was considered high for ten studies [18,19,20,21,22,23,24,25,26,27] and moderate for one study [17].

## 4. Discussion

This literature review identified 11 studies about vitamin D and COVID-19 infection in subjects aged 60 years or older. Seven articles [19,20,21,23,24,25,26] investigated the link between vitamin D deficiency and COVID-19 infection in subjects aged 60 years or older. These seven studies compared mortality and/or risk of adverse outcomes in COVID-19-positive patients according to vitamin D levels. In two of these seven studies [19,26], serum vitamin D level was higher in COVID-19-negative patients. Four studies investigated vitamin D supplementation during the acute phase of COVID-19 infection [17,18,22,27].

Concerning the relationship between vitamin D levels and comparison of COVID-19 infection status, in the study by Baktash et al. [19], the median vitamin D level was low, regardless of COVID-19 status (less than 30 ng/mL). Similar observations were made in the younger adult population. Meltzer et al. [28] compared the COVID-19 status of 489 individuals (mean age 49.2 ± 18.4 years) according to their vitamin D levels, and concluded that positive COVID-19 status was statistically associated with vitamin D deficiency (relative risk= 1.77, *p* = 0.02). In a population of adults (mean age 52.3 ± 20.5), Im et al. [29] also reported lower vitamin D levels in COVID-19-positive patients compared with a COVID-19-negative group (*p* < 0.0001). COVID-19-positive subjects were statistically more likely to be vitamin D deficient (*p* = 0.003) or insufficient (*p* = 0.001) than COVID-19-negative subjects. This suggests that the lower the vitamin D level, the more likely the subjects are to develop COVID-19 infection. There was a high prevalence of low vitamin D levels, even in countries with abundant sunshine, particularly in aged people [30]. Observational studies suggest an association between low serum vitamin D level and susceptibility to acute respiratory tract infection [31].

Concerning the relationship between vitamin D level and outcomes in COVID-19, patients with vitamin D deficiency have worse clinical outcomes than non-deficient patients in terms of mortality [19,20,21,23,24,25,26]. Brenner et al. [32] showed similar results in a cohort of older adults with respiratory diseases during a median of 15.3 years of follow-up. Mortality was consistently higher among participants with vitamin D deficiency. Moreover, there was a significant survival difference concerning respiratory disease mortality between the group with vitamin D deficiency and sufficient vitamin D (*p* < 0.0001), as well as for the comparison of subjects with vitamin D insufficiency and sufficient vitamin D (*p* = 0.023). Ye et al. [33] examined the relationship between vitamin D levels and clinical characteristics, and severity of COVID-19 infection in an adult population. They showed a higher rate of vitamin D deficiency in severe/critical COVID-19 cases. Maghbooli et al. [34] published similar results. The severe form of COVID-19 infection appears preferentially beyond the seventh day after the onset of symptoms [1,2,4]. Bacterial co-infection (including pneumonia secondary to COVID-19 infection) is infrequent in hospitalized patients with COVID-19 infection, as observed by Wang et al. [35]. They concerned less than 3% of patients in their retrospective observational cohort study of 1396 patients with COVID-19 infection. In severe forms, patients with COVID-19 infection develop acute respiratory failure associated with a severe inflammatory syndrome [4]. This is a new form of viral pneumonia with typical findings in chest CT images (multi-lobular ground-glass opacities, bilateral and multi-lobular involvement, peripheral distribution) [2]. Bacterial pneumonia produces consolidation, interlobular reticular opacities, and centrilobular nodules [36]. Acute respiratory distress syndrome in COVID-19 infection is thought to be secondary to dysregulation of the immune system. Several authors have highlighted abnormally high levels of proinflammatory cytokines in patients developing severe forms of COVID-19 infection with pulmonary involvement and acute respiratory distress [9,10,37]. Some authors have suggested that vitamin D could be active on the immune and respiratory systems [38], notably by modulating cytokine production [13]. Additionally, vitamin D deficiency may decrease the immune system’s defenses against COVID-19 [29]. However, the effects of vitamin D metabolites in the modulation of immune response to respiratory viruses remain unclear [39].

Aged people are at greater risk of severe forms of COVID-19 infection [7] and at risk of vitamin D deficiency [32,40], prompting some authors to suggest that vitamin D supplementation may improve the prognosis of aged people infected by SARS-CoV-2 [17,18,27]. In our review, four articles concerned vitamin D supplementation in patients with COVID-19 infection [17,18,22,27]. Results suggest that older people with vitamin D supplementation during the acute phase of COVID-19 infection were at lower risk of adverse outcomes (mortality, high flow oxygen therapy needs, or ICU support). Entrenas Castillo et al. [41] showed similar findings in younger adults with COVID-19 infection. Ginde et al. [42] showed that monthly vitamin D supplementation reduced the incidence of acute respiratory infection in older people in long-term care units.

While vitamin D supplementation’s positive effect seems to be established, it remains unclear what the ideal supplementation regimen is (dose, frequency of administration, duration). In a consensus statement on vitamin D research, Giustina et al. [40] published target thresholds to limit certain risks related to vitamin D deficiency (for skeletal and non-skeletal health). However, there is currently no consensus on the level of vitamin D needed to limit the immune system’s risk in general and the risk of COVID-19 infection in particular. Ye et al. [33] suggested that the level of vitamin D required to limit the risk of adverse effects in COVID would be greater than 16.5 ng/mL. In our review, supplementation regimens differed between studies, precluding comparison. Annweiler et al. [18] showed that vitamin D supplementation’s protective effect is greater in patients with chronic supplementation. Some authors [18] observed that vitamin D supplementation (80,000 IU vitamin D3) initiated after COVID-19 diagnosis was not associated with any beneficial effect compared to no vitamin D supplementation in older patients with acute COVID-19 infection. Chronic supplementation instinctively seems more suitable for achieving both short- and long-term effects on the immune system. Martineau et al. [31] concluded that vitamin D protected patients against respiratory infections, especially if patients were deficient or received daily or weekly supplementation. Bolus doses seemed to be less effective. If vitamin D deficiency can be rapidly corrected by the ingestion of 50,000 IU once a week, and given that adverse effects are rare [43], some data suggest that daily or weekly supplementation may be more effective for effect on immunity. Bolus doses or routine supplementation seemed to be less effective [44]. A randomized trial would provide a clearer conclusion on the value of daily, weekly, or monthly vitamin D supplementation in preventing respiratory viral infections. Moreover, it would make it possible to define the interest of a supplementation, whatever the initial level of vitamin D of the patients.

Our study has some limitations. The articles included in this review reported vitamin D levels and compared populations according to the presence or absence of vitamin D deficiency. First, there is often a lack of information on the samples' timing used to measure vitamin D, especially in observational studies. If they were at the admission for COVID-19, these results could not be interpreted to exclude reverse causality. Since vitamin D is involved in the immune response, its level is likely to vary according to the immune response [39]. Inflammation may reduce 25OHD metabolism resulting in reduced circulating levels [21]. Second, there is no consensus on the exact definition of vitamin D deficiency and/or insufficiency. Consequently, thresholds differed across studies, rendering comparisons difficult. Nevertheless, despite the different cutoff values used, all studies concur in concluding that vitamin D deficiency is deleterious, and all cutoff values remained within a relatively small range of each. Furthermore, the number of subjects included in the individual studies was limited (ranging from 20 to 185). For the studies investigating vitamin D supplementation, in particular, the number of subjects in each study arm was small. In addition, the studies were observational, thus yielding a lower grade of evidence. It would have been interesting to conduct a meta-analysis of the four studies of vitamin D supplementation; however, this could not be done because the four studies did not have the same endpoint. Conversely, all studies were evaluated for methodological quality using the NOS, and 10 out of 11 were found to be of high quality, and only one of moderate quality (none was poor quality). Finally, the influence of changes in renal function (acute or chronic) on vitamin D levels is not often discussed in the different articles. Chronic kidney disease reduces the vitamin D hydroxylation process and the formation of active metabolites. Chronic kidney disease is associated with a significantly greater risk of mortality in COVID-19 [45]. Acute kidney injury is frequent in COVID-19 infection and is associated with poor outcomes [46]. Further studies are needed to confirm the link between vitamin D deficiency, kidney disease, and COVID-19 infection.

Our study also has some strengths. This is the first systematic review of the literature regarding using vitamin D and COVID-19 infection in a population of older adults. All the authors of the articles included were contacted individually to recover missing data that was not in the publications and to verify the accuracy of the published results. The articles included in this review were analyzed by senior researchers with many years of experience in geriatrics, infectious diseases and epidemiology.

In conclusion, the data from this systematic review of the literature argue in favor of an association between vitamin D deficiency and an increased risk of infection with COVID-19 in patients aged 60 years and older. In addition, vitamin D deficiency seems to expose older subjects to an increased risk of unfavorable disease course and outcomes in case of COVID-19 infections than patients with adequate vitamin D levels. The lower the level of vitamin D, the higher risk of severe forms of infection and death. Chronic vitamin D supplementation is already indicated in primary prevention for certain skeletal and non-skeletal pathologies and should be integrated into the preventive strategies for certain viral diseases and for COVID-19. Supplementation with vitamin D is easy to implement, as vitamin D is available as an oral solution. Adverse effects of vitamin D supplementation are rare, and the therapeutic target is wide and requires no biological surveillance. All these arguments plead for using vitamin D supplementation as a simple prevention strategy in respiratory infections in general and COVID-19 in particular.

## Figures and Tables

**Figure 1 nutrients-13-01339-f001:**
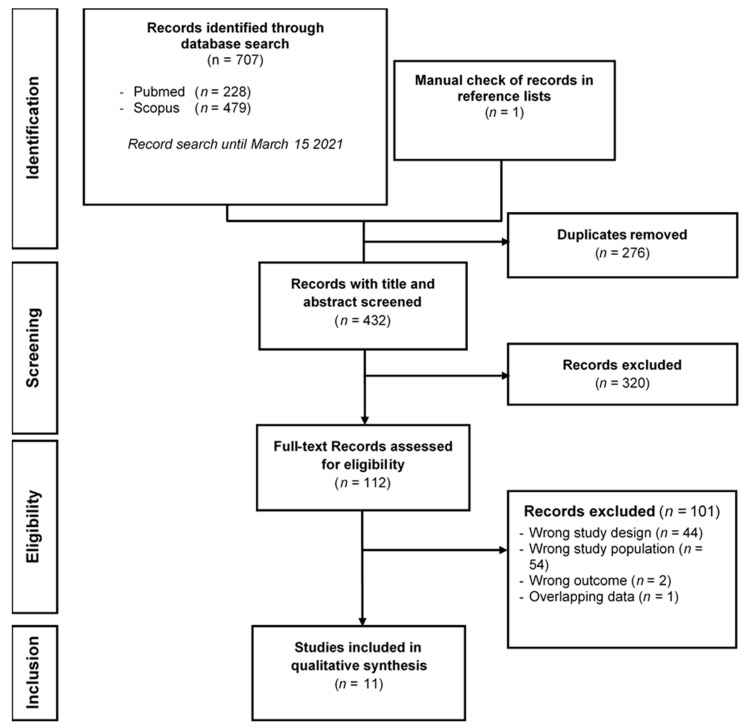
PRISMA (Preferred Reporting Items for Systematic Reviews and Meta-Analyses) flow diagram of the records included in the systematic review.

**Table 1 nutrients-13-01339-t001:** Description of the 11 studies included in the systematic review.

Author, Year	Setting	Study Design	N	Female Sex	Age (Years)	COVID-19+ Patients	Comparison
Annweiler C, 2020 [17]	NH	Retrospective cohort	66	77.3%	88 ± 9 *	100%	Supplementation
Annweiler G, 2020 [18]	ACU	Retrospective cohort	77	49.4%	88 ± 5 *	100%	Supplementation
Giannini S, 2021 [22]	ED	Retrospective cohort	77	49.4%	78 ± 10 *	100%	Supplementation
Tan CW, 2020 [27]	Hospital	Prospective cohort	20	50.0%	66 ± 4 *	100%	Supplementation
Baktash V, 2020 [19]	ED	Prospective cohort	105	45.7%	81 (65–102) ^†^	67%	Deficiency
Carpagnano GE, 2020 [20]	ICU	Retrospective cohort	27	29.6%	72 ± 9 *	100%	Deficiency
Cereda E, 2020 [21]	Hospital	Prospective cohort	106	49.0%	78 ± 9 *	100%	Deficiency
Hars M, 2020 [23]	ACU	Retrospective cohort	160	59.4%	86 ± 7 *	100%	Deficiency
Macaya F, 2020 [24]	ED	Retrospective cohort	55	56.4%	75 ± 10 *	100%	Deficiency
Radujkovic A, 2020 [25]	ACU and community	Prospective cohort	185	42.3%	71 ± 8 *	100%	Deficiency
Sulli A, 2021 [26]	Hospital and community	Retrospective case control	130	53.8%	76 ± 13 *	50%	Deficiency

NH: nursing home; ACU: Acute care unit; ICU: intensive care unit; ED: emergency department. Age: * mean ± standard deviation; ^†^ median (range).

**Table 2 nutrients-13-01339-t002:** Description of outcome criteria and results for the association between vitamin D supplementation and COVID-19 in elderly patients.

Author, Year	N	Supplementation		Outcome		Results		Death
		Products	Regimen	Primary	Secondary	Primary	Secondary	
Annweiler C, 2020 [17]	66	Vitamin D3	Group 1: oral bolus of 80 kIU in the week following suspicion or diagnosis of COVID-19	Death during follow-up	Severe COVID-19 (OSCI score ≥5)	Group 1: 17.5%	Group 1: 21.1%	22.7%
			Group 2: no supplementation			Group 2: 55.6%	Group 2: 66.7%	
Annweiler G, 2020 [18]	77	Vitamin D3	Group 1: oral bolus of 50 kIU per month, or 80 or 100 kIU every 2–3 months over the preceding year	14-day death	Severe COVID-19 (OSCI score ≥5)	Group 1: 6.9%	Group 1: 10.3%	19.5%
			Group 2: single oral bolus of 80 kIU within a few hours after COVID-19 diagnosis			Group 2: 18.8%	Group 2: 25.0%	
			Group 3: No supplementation			Group 3: 31.3%	Group 3: 31.3%	
Giannini S, 2021 [22]	77	Vitamin D3	Group 1: oral 400 kIU vitamin D (2*100 kIU daily for two consecutive days)	Death and/or ICU support	Death	Group 1: 43.3%	Group 1: 33.3%	28.6%
			Group 2: no supplementation			Group 2: 57.4%	Group 2: 25.5%	
Tan CW, 2020 [27]	20	Vitamin D3, B12, magnesium,	Group 1: single daily dose 1 kIU for ≤14 days	Oxygen therapy requirement and/or ICU support	Oxygen therapy requirement but no ICU support	Group 1: 25.0%	Group 1: 12.5%	0.0%
			Group 2: no supplementation			Group 2: 58.3%	Group 2: 16.7%	

OSCI: ordinal scale for clinical improvement. The OSCI is the 9-point World Health Organization ordinal scale rating clinical improvement in COVID-19. It distinguishes several severity levels with a score ranging from 0 (no clinical or virological evidence of infection) to 8 (death).

**Table 3 nutrients-13-01339-t003:** Description of vitamin D status in elderly patients with and/or without COVID-19.

Author, Year	N	Serum Vitamin D Level (ng/mL)	Serum Vitamin D Level in COVID-19+ Patients (ng/mL)	Serum Vitamin D Level in COVID-19- Patients (ng/mL)	Subjects with Vitamin D Deficiency
Baktash V, 2020 [19]	105	14.3 ± *	10.8 ± 8.8 ^¶^	20.8 ± 16.0 ^¶^	45.7%
Carpagnano GE, 2020 [20]	27	16.1 ± 14.0 ^†^	16.1 ± 14.0 ^†^	NA	37.0%
Cereda E, 2020 [21]	106	13.9 ± 11.7	13.9 ± 11.7	NA	74.5%
Hars M, 2020 [23]	160	24.0 ± 15.2 ^¶^	24.0 ± 15.2 ^¶^	NA	36.9%
Macaya F, 2020 [24]	55	17.0 ± 22.0 ^¶^	16.7 ± 22.0 ^¶^	NA	52.7%
Radujkovic A, 2020 [25]	185	19.7 ± 12.4 ^†^	19.7 ± 12.4 ^†^	NA	63.7%
Sulli A, 2021 [26]	130	12.1 ± 17.0 ^¶^	7.9 ± 15.0 ^¶^	16.3 ± 19.0 ^¶^	83.8%

* standard deviation not available. ^†^ Mean ± standard deviation. ^¶^ Median ± interquartile range. NA: not appropriate.

**Table 4 nutrients-13-01339-t004:** Description of outcomes and results for the association between vitamin D status and COVID-19 in elderly patients.

Author, Year	N	Outcome		Results						
		Primary	Secondary	Primary			Secondary			Deaths
				Overall	Deficiency	No Deficiency	Overall	Deficiency	No Deficiency	
Baktash V, 2020 [19]	105	In-hospital death ^‡^	Composite ^‡,§^	14.3%	15.4%	12.9%	48.6%	59.0%	35.5%	14.3%
Carpagnano GE, 2020 [20]	42	10-day death		18.5%	30.0%	11.8%				18.5%
Cereda E, 2020 [21]	106	Prevalence of deficit	Composite ^||^	74.5%	100.0%	0.0%	90.6%	92.4%	85.2%	31.1%
Hars M, 2020 [23]	160	In-hospital death		25.0%	32.2%	20.8%				25.0%
Macaya F, 2020 [24]	80	Composite ^¶^		43.6%	44.8%	42.34%				21.3%
Radujkovic A, 2020 [25]	97	IMV and/or death	Death	24.7%	57.7%	12.7%	15.5%	46.2%	4.2%	15.5%
Sulli A, 2021 [26]	65 ^#^	In-hospital death		15.4%	17.6%	7.1%				15.4%

^‡^: Results are only for COVID-19-positive patients (*n* = 70). ^§^: non-invasive ventilation support and admission to high dependency unit, COVID-19 radiographic changes on chest X-ray. ^||^: in-hospital mortality, ICU admission, severe pneumonia. ^¶^: death, ICU admission, need for higher oxygen flow. IMV: invasive mechanical ventilation. ^#^ Only COVID-19-positive patients are considered in this table.

**Table 5 nutrients-13-01339-t005:** Study quality assessment using Newcastle–Ottawa scale (NOS).

Study First Author, Month Year	Selection	Comparability	Outcome	Total Score	Quality Rating
Annweiler C, September 2020 [17]	**	*	***	6	Moderate
Annweiler G, November 2020 [18]	***	*	***	7	High
Giannini S, January 2021 [22]	***	**	***	8	High
Tan CW, December 2020 [27]	****	**	***	9	High
Baktash V, August 2020 [19]	****	*	***	8	High
Carpagnano GE, August 2020 [20]	****	*	***	8	High
Cereda E, October 2020 [21]	****	**	***	9	High
Hars M, October 2020 [23]	****	**	***	9	High
Macaya F, October 2020 [24]	****	**	***	9	High
Radujkovic A, September 2020 [25]	***	*	***	7	High
Sulli A, February 2021 [26]	****	**	***	9	High

NOS scores of ≥7 were considered as high-quality studies and of 5–6 as moderate quality. * = 1 point in the NOS score (e.g., ** for selection means 2 points in the NOS score).

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
