# Peer review of "Relation between Vitamin D and COVID-19 in Aged People: A Systematic Review"

_nutrients, 2021, doi:10.3390/nu13041339_

Round 1

Reviewer 1 Report

Here, the authors describe a systematic review of 3 vitamin D supplementation quasi-experimental studies and 9 studies of various designs measuring serum 25(OH)D levels, assessing the relationships of these with COVID-19 severity outcomes. These 12 studies found that those on vitamin D supplements had better clinical outcomes than did those not supplemented, while the studies of vitamin D deficiency found those who were deficient had worse COVID-19 outcomes. The authors conclude from these data that vitamin D supplementation may be useful for moderating COVID-19 and other respiratory infections.

While I appreciate the evidence from various studies that vitamin D supplementation may be beneficial for respiratory infection prevention and moderation, and particularly there being appreciable biological plausibility for vitamin D having a modulating role in innate and adaptive immunity, the conclusions of this systematic review are too strong. For respiratory infections in general, meta-analyses of randomised controlled trial results have suggested a beneficial effect among persons starting out as deficient (see Martineau BMJ 2017). However, the studies included in this systematic review are of varied quality, despite the NOS scores given by the authors all being quite high, and in any event their study designs preclude definitive causal inference, this requiring RCT. Thus, while these results are promising and suggestive, it is too premature to suggest anything as far as translation to practice. At most, these results suggest that further observational studies and RCTs might be undertaken. As the authors note, the included studies are of small sample size and only three are prospective in design. There are also other issues which I’ll go into.

There is a statement the authors make in the Abstract and the Introduction that “vitamin D regulates the immune response by influencing viral load and suppressing cytokine production”. This statement is partly wrong and partly too general. Vitamin D does not regulate the immune response by reducing viral load, this is nonsensical. Also, it doesn't suppress cytokine production, it induces a less pro-inflammatory state and this vicariously reduces those cytokines but it doesn't act upon cytokine production directly. There is a more correct statement in the Introduction about vitamin D activating innate immunity and reducing overactive adaptive immune response. Use this instead.

A significant issue lies in the vitamin D deficiency study as it’s unclear when the samples used for 25(OH)D measures were taken. If they were at the admission for COVID-19, these results cannot be interpreted to exclude reverse causality. Indeed, despite some of these studies being prospective or retrospective in nature, it’s probable that this period is measured in days in which case it’s effectively contemporaneous as far as 25(OH)D production and metabolism is concerned. Thus, I’m not sure that these studies really inform the literature as far as a potential causal relationship. At least the supplementation studies have some material causal window, either pre-COVID supplement use or a bolus treatment or such like, which would potentially allow some inference of causal relationships, despite the lack of randomisation. The serum 25(OH)D studies, though, not so much.

The Introduction doesn’t give a justification for why the focus is on persons aged 60+. Though there is passing mention that COVID-19 severity is worse in the aged, it’s unclear from the Introduction as to why 60+ is the focus.

Further to this, in looking at the SD and IQR of the ages in Table 1, it would seem a few of these would include patients below age 60, which would contradict the study aims. Were those people somehow excluded or how are these studies able to be included if the remit is only 60+?

Regarding the excluded studies on lines 128-129, can some detail as to what these entail be provided? What would be the wrong study design? Does wrong study population mean <60yo? What is wrong outcome? 

The D’Avolio and the Raisi-Estabragh study would seem problematic if their measure is vitamin D deficiency but they don't define what that is. Given as these studies are all “Not applicable” for the results in Table 4, I would suggest excluding these studies.

Similarly, I would suggest excluding the Goncalves study as its primary and secondary outcomes have nothing to do with COVID-19, being obesity and hypovitaminosis D.

For Tables 3 and 4, there is no indication whether the intergroup associations differ significantly. Could p-values and/or 95% CIs be added?

Other:

  • Minor point but COVID-19 should be capitalized like that.
  • The Abstract needs to be clear that the included studies were all observational, not interventional.
  • In the Abstract, it would help giving some better detail as to what “better clinical outcomes” entails, either as a list or a “such as”.
  • Throughout, be clear it’s activated vitamin D, not just vitamin D, when talking about immunomodulatory effects.
  • In the Introduction, be specific when you refer to lowering interleukin levels (lines 60-61).
  • Note that formatting of ref 13 on line 63 is not right.
  • In the Methods, please specify why a meta-analysis was not done. Presumably due to heterogeneity of the included studies but just state why.
  • In Table 1, suggest you move the Tan supplementation study up with the other two supplementation studies.
  • Given as Table 4's results just relate to deficiency, Deficiency in Table 2 would seem the only column needed? Perhaps these values could just be provided in the Table 4 subcaption?

Author Response

Dear Reviewer,

Please find enclosed the file addressing your different comments on our manuscript.

Best regards

Reviewer 2 Report

Moustapha Dramé et al. investigated the relation between vitamin D and Covid-19 in aged people: A systematic review. The study is interesting and may have a clinical impact, but several study concerns should be mentioned. 1. A literature search was performed in PubMed© and Scopus© for all publications from inception up to 5 November, 2020. In total, 517 studies were identified, of which 12 were included in the final review. I strongly recommended to extend the due time to Feb, 2021. The final results could be more powerful and adequate.   2. The discussion seems to be unsatisfactory and simple. Multiple clinical impacts such as acute kidney injury, chronic kidney disease….., and the association between vitamin D and Covid-19 should be also discussed more in your review .   3. Finally, what is the difference between virus infections associated pneumonia or bacteremia related pneumonia and Covid-19 should be also to make a brief discussion.

Author Response

(The authors gave the same response as above.)

Reviewer 3 Report

see enclosed file

Author Response

(The authors gave the same response as above.)

Round 2

Reviewer 1 Report

Thanks to the authors for their responses and edits from my comments. The article is much improved. 

Reviewer 2 Report

All comments have been addressed adequately. I have no further suggestion. I believe that the authors have improved the understanding of their work for the readers of this article.

Reviewer 3 Report

The authors addressed all my concerns